# Performance Enhancement of Pentacene-Based Organic Thin-Film Transistors Using a High-K PVA/Low-K PVP Bilayer as the Gate Insulator

**DOI:** 10.3390/polym13223941

**Published:** 2021-11-15

**Authors:** Ching-Lin Fan, Hou-Yen Tsao, Yu-Shien Shiah, Che-Wei Yao, Po-Wei Cheng

**Affiliations:** 1Graduate Institute of Electro-Optical Engineering, National Taiwan University of Science and Technology, 43 Sec. 4, Keelung Road, Taipei 106, Taiwan; ohye0220@gmail.com (H.-Y.T.); joshshiah@gmail.com (Y.-S.S.); ja5191111@gmail.com (C.-W.Y.); b380424@gmail.com (P.-W.C.); 2Department of Electronic and Computer Engineering, National Taiwan University of Science and Technology, 43 Sec. 4, Keelung Road, Taipei 106, Taiwan

**Keywords:** organic TFT, pentacene, gate dielectric, high-K, field-effect mobility, surface morphology

## Abstract

In this study, we proposed using the high-K polyvinyl alcohol (PVA)/low-K poly-4-vinylphenol (PVP) bilayer structure as the gate insulator to improve the performance of a pentacene-based organic thin-film transistor. The dielectric constant of the optimal high-K PVA/low-K PVP bilayer was 5.6, which was higher than that of the single PVP layer. It resulted in an increase in the gate capacitance and an increased drain current. The surface morphology of the bilayer gate dielectric could be suitable for pentacene grain growth because the PVP layer was deposited above the organic PVA surface, thereby replacing the inorganic surface of the ITO gate electrode. The device performances were significantly improved by using the bilayer gate dielectric based upon the high-K characteristics of the PVA layer and the enlargement of the pentacene grain. Notably, the field-effect mobility was increased from 0.16 to 1.12 cm^2^/(Vs), 7 times higher than that of the control sample.

## 1. Introduction

Pentacene-based organic thin-film transistors (OTFTs) have recently attracted much attention because of their potential for use in flexible displays, large-area chemical sensors for artificial skin applications, and radio-frequency power transmission devices [1,2,3,4,5]. Conventionally, the pentacene channel combined with poly-4-vinylphenol (PVP) as the gate insulator has been recognized as the most adequate construction for OTFTs [6]. However, the relatively low dielectric constant (low-K) of PVP may necessitate excessive power consumption in order to achieve sufficient operational capability [7]. To overcome this obstacle requires either increasing the gate dielectric capacitance with the reduced thickness of the dielectric or fabricating OTFTs with high dielectric constants (high-K) [8,9]. However, the thickness reduction of the organic dielectric rapidly increases the defects and pinholes in the dielectric, leading to degradation of the device. Low polymer-based dielectric thicknesses can induce pinholes as the substrate coverage by dielectric layers may not be homogeneous upon their deposition [10]. Accordingly, the adoption of high-K material appears to be a better solution [11,12]. The double-layer dielectric made of yttrium oxide and a PVP layer may provide a better combination [13]. Due to their flexible applicability and excellent film growth properties [14], pentacene thin-film transistors were fabricated and characterized with PVA thin films used as a gate dielectric [15]. Similar to the PVP dielectric, the organic PVA is a polar polymer with abundant hydroxyl –OH groups; however, the natural hydrophilicity of this polymer may result in increased difficulty when using a pentacene film on a PVA surface [16]. Therefore, an appropriate curing procedure for cross-linking the –OH group should be introduced to the sequential fabrication process to eliminate –OH groups of the PVA and enhance the grain growth of the pentacene film. From the perspective of curing, the use of various cross-linking agents including dichromate [17], boric acid [18], and ammonium bicarbonate [19] have been reported in a number of earlier research papers. Nevertheless, most of the reported cross-linking agents possess highly toxic characteristics likely to cause significant damage to the human body. As a result, double-stacked insulators consisting of a high-K PVA layer without cross-linking, combined with a cross-linking low-K PVP layer, can be used. The previously reported cross-linking agents are not only highly toxic to humans but also require an additional process for the cross-linking step. As a result, our proposed double-stacked insulators with high-K PVA/low-K PVP can be conducted to overcome these issues.

In this study, we used a low-K PVP layer above a high-K PVA layer as the bilayer gate dielectric (high-K PVA/low-K PVP) to facilitate the grain growth of a pentacene film. Consequently, the performance of devices is improved by using the hydrophobic PVP layer and a PVA layer with high-K characteristics. In addition, the surface morphology of the bilayer gate dielectric (high-K PVA/low-K PVP) allows more suitable growth of the pentacene grain because the PVP layer is deposited above the organic PVA surface instead of an inorganic ITO gate surface. Compared with other similar papers, the improved u_FE_ in our study is about 1.12 cm^2^/Vs, significantly better than that of the reported papers previously [19,20,21,22]. The obvious performance improvement can be attributed to the high-K PVA/low-K PVP bilayer structure based upon the high-K characteristics of PVA and the hydrophobic surface of PVP. This led to an increased drain current and an enlarged pentacene grain size, which in turn resulted in improved performances. Thus, it is believed that the proposed high-K PVA/low-K PVP structure is a good candidate for performance improvement because it can not only improve the device performances but also provide the advantages of a simple process, low cost, and the avoidance of the cross-linking process of PVA using toxic agents, in comparison with similar reports [17,18,19,20,21,22].

## 2. Materials and Methods

The glass substrate with an indium tin oxide (ITO resistivity: 20–40 Ω·cm) layer was prepared as a gate electrode of the bottom-gate top-contact device. The sequential PVA and PVP dielectric layers were spin-coated on the ITO glass. For the first PVA dielectric layer, we dissolved PVA (molecular weight = 46,000–186,000) in different weight percentages (25, 16, and 12 wt%) and baked these in a vacuum oven at 130°C for 1 h to reduce the –OH groups. For the second PVP layer, PVP powder was mixed with poly (melamine-co-formaldehyde) methylated (PMF) in the propylene-glycol-monomethyl-ether-acetate (PGMEA) solvent, which then went through a cross-linking procedure in a vacuum oven at 180 °C for 1 h to manufacture the PVP layer (PVP/PMCF/PGMEA = 2:1:20). Next, a shadow mask patterned a 50 nm thick pentacene (Aldrich Chem. Co., Milwaukee, WI, USA, 99% purity) layer, which was deposited onto the dielectric layer by vacuum thermal evaporation. The evaporation rate was 0.1 A°/s without the additional substrate heating. Finally, silver source/drain electrodes were deposited by thermal evaporation. Figure 1a,b indicates the cross-section structure of the fabricated OTFT with a high-K PVA/low-K PVP bilayer gate dielectric and a PVA or PVP single gate dielectric. Control samples were also fabricated using a single dielectric layer of PVA or PVP, respectively, and metal–insulator–metal (MIM) capacitors, which compared capacitance measurements.

All devices were measured through a semiconductor parameter analyzer (HP 4145B). The thickness was calculated using a scanning electron microscope (SEM, JEOL JSM-6390 LV, Stoneridge Drive, CA, USA), and the capacitance value was recorded by an LCR meter (WK4100 Series, Taipei, Taiwan). Use of the Fourier transform infrared method (FTIR, Astex PDS-17 system, Mass., USA) and an atomic force microscope (AFM, Veeco Dimension 5000 scanning probe microscope, Lise-Meitner, Germany) provided the mechanistic explanation of the improved device.

## 3. Results and Discussion

To conduct the proposed idea, we first needed to examine the thickness of the dielectric layer since capacitance was strongly correlated with the thickness of the dielectric, based on Equations (1) and (2):C = (ε_0_ kA)/t(1)
C_Total_ = 1/[(1/C_PVA_) + (1/C_PVP_)](2)
where C is the capacitance of the gate insulator layer, A is the electrode area, ε_0_ is the vacuum dielectric constant, and t is the thickness of the dielectric layer. Generally, the adhesion of the PVA solution was controlled by the PVA concentration, the main factor for the thickness of PVA under the same rotation speed. Thus, we prepared different concentrations of PVA solution to acquire an optimal value for the capacitance of PVA. The different thicknesses with the various PVA concentrations, including 25, 16, and 12 wt%, were estimated to be 1380 nm, 510 nm, and 300 nm, respectively, as shown in Figure 2. As previously mentioned, organic dielectric thickness suffers from severe performance degradation through the thickness reduction process and then induces complexity to the device [23]. Therefore, we fixed the thickness of the second PVP dielectric layer at about 500 nm, which was a reasonable value for OTFT fabrication. 

Figure 3 shows the capacitance for different concentrations of PVA combined with PVP of 500 nm. It demonstrates the negative correlation between film thickness and capacitance. The frequency of capacitance measurement was 1 kHz. The dielectric constant of the high-K PVA/low-K PVP bilayer with the PVA concentration at 12 wt% using Equations (1) and (2) was calculated to be 5.6. The calculated dielectric constant of the high-K PVA/low-K PVP bilayer structure was smaller than that of the single PVA layer, which was reported to be 9.2 [24] and was larger than that of the single PVP layer, which was reported to be about 3.5 [25]. Thus, the proposed high-K PVA/low-K PVP bilayer structure could obviously increase the effective capacitance in comparison to that attained with the single PVP layer, resulting in an increased drain current [11,12].

The hydroxyl (–OH) groups are known to be preferentially bonded with those in PMF through the curing step, which is called a cross-linking process. Therefore, the cross-linking efficiency can be evaluated by Fourier transform infrared (FTIR) to measure the number of –OH groups. Figure 4 shows the FTIR measurement of the bilayer gate dielectric with various PVA concentrations combined with the PVP layer of 500 nm. The –OH peaks can be observed at around 3200 cm^−1^–3500 cm^−1^. It is found that the number of –OH groups can be obviously reduced for thinner dielectric films with PVA (12 wt%)/PVP, as shown in Figure 4. The reduction of –OH groups might be due to fewer –OH groups within the thinner dielectric films, which led to more efficient –OH elimination through the baking process [26]. Thus, the PVA concentration of 12 wt% provided the most suitable parameters in our study. 

Figure 5 shows the transfer characteristics (I_DS_-V_GS_) of the OTFT with the PVA (12 wt%)/PVP bilayer gate insulator, single PVA gate layer, and single PVP gate layer, all of which were measured at a drain voltage (V_DS_) of −20 V. Figure 5b shows the gate leakage current of the device with a high-K PVA/low-K PVP bilayer is significantly decreased by about four orders of magnitude than that of the device with the single PVA structure. Additionally, the gate current with a high-K PVA/low-K PVP bilayer is comparable to that with a single PVP layer. Figure 5c,d shows the output curves (I_DS_–V_DS_) of the devices with high-K PVA/low-K PVP and PVP dielectrics, respectively, as a function of drain/source voltage (V_DS_) for gate/source voltages (V_GS_) of 0, −10, −20, and −30 V. As a result, the output current (I_DS_) of the devices with a high-K PVA/low-K PVP bilayer gate insulator is obviously larger than that of the devices with PVP dielectric layer. Thus, the proposed scheme with a high-K PVA/low-K PVP bilayer as a gate insulator will be a good candidate, which is not only for improving the electrical characteristics of the pentacene-based OTFTs but also for acting as a good gate insulator with reduced gate leakage current. The field-effect mobility and threshold voltage were calculated in the saturation region by fitting the |I_DS_|^1/2^ curve based on Equation (3): I_DS_ = (1/2μ_FE_C_i_W/L)(V_GS_ − V_TH_)^2^(3)
where μ_FE_ is the field-effect mobility, C_i_ is the capacitance density of the gate insulator, V_TH_ is the threshold voltage, and W (width) and L (length) are the dimensions of the semiconductor channel defined by the source and drain electrodes. 

The maximum and minimum values of drain current measured at a drain voltage of –20 V were designated as I_ON_ (on-current) and I_OFF_ (off-current), respectively. It was found that the device performance was significantly improved by the use of the high-K PVA/low-K PVP bilayer gate dielectric. The total electrical parameters are listed in Table 1. In comparison to the conventional device with a PVP dielectric layer, the field-effect mobility (μ_FE_) of the device with a high-K PVA/low-K PVP bilayer dielectric layer was significantly increased from 0.16 to 1.12 cm^2^/(Vs). In addition, the threshold voltage (V_TH_) and I_ON_/I_OFF_ ratio were also obviously improved. It is believed that the increased gate dielectric constant is one of the main factors contributing to performance improvement. The large gate capacitance can result in more charge per area unit in the channel region for a given gate bias [11,12]. We presumed that the grain size of pentacene would also contribute to improved mobility.

It was also found that the device with a single PVA dielectric did not show the switching characteristics of a semiconductor. Because of that, the hydrophilic surface of PVA results in poor surface conditions, which inhibits the growth of the pentacene grain. It has been reported [27] that the hydrophilic condition can be represented by the measured contact angle of the surface. The contact angle of the samples with a single PVA, single PVP, and bilayer high-K PVA/low-K PVP, respectively, were measured using the sessile drop method, as shown in Figure 6. The single PVA sample presented a low contact angle of 35.92°, which represented the hydrophilic surface. Yu et al. [28] proposed that the hydrophilic surface inhibits the growth of pentacene causing a small grain size, resulting in high-density grain boundary defects, leading to poor carrier transportation. The contact angles of the single PVP and high-K PVA/low-K PVP surfaces were similar and comparable at 69.03° and 66.54°. The contact angle of the single PVP and high-K PVA/low-K PVP surfaces showed greater hydrophobic activity than that of the single PVA surface. Table 2 shows the contact angles of DI water and diiodomethane and the surface energies of gate insulators. The γ^d^ and γ^p^ represent dispersion and polar components of the surface energy (γ), respectively. It shows the surface energy values, 72.57 mJ/cm^2^, 52.95 mJ/cm^2^, and 50.90 mJ/cm^2^ for High-K PVA, high-K PVA/low-K PVP, and low-K PVP, respectively. The decreased surface energy of the PVA/PVP bilayer is comparable to that of the PVP layer due to the removal of –OH groups from the PVP by the cross-linking process. It is noted that the surface energy of PVA is higher than that of PVA/PVP or PVP layers due to lots of –OH groups. It is also consistent with the measured contact angles in Figure 6 because the smaller contact angle represents the higher surface energy. The smaller contact angle and higher surface energy of PVA show the hydrophilic surface with OH groups, which could result in a stronger interaction between the dielectric surface and the pentacene molecules during the deposition to inhibit the growth of the pentacene grain. In contrast, pentacene exhibits large grains when grown on the hydrophobic insulator surface with a high contact angle and low surface energy.

Figure 7 shows that the PVA layer has the smallest grain size and is also consistent with the smallest contact angle. The average grain sizes of pentacene were 1.58 and 2.16 µm for the PVP dielectric layer and the high-K PVA/low-K PVP bilayer, respectively. Since the PVP layer of the high-K PVA/low-K PVP bilayer was deposited onto the organic PVA layer instead of the inorganic ITO surface as shown in Figure 1a,b, it was presumed that the surface morphology of the high-K PVA/low-K PVP bilayer gate dielectric was more suitable than that of the PVP single layer. Thus, the more suitable surface morphology of the high-K PVA/low-K PVP bilayer could significantly increase the pentacene grain size, which also caused the obviously improved μ_FE_ of the device with a high-K PVA/low-K PVP bilayer. It is well known that there are numerous traps in the grain boundary of the polycrystalline pentacene film. The film with the larger grain size has a reduced amount of grain boundary, resulting in a reduced number of traps in the film. In regard to the correlation between mobility and grain size, many reports have described the mechanism [29,30]. Matsubara et al. found that carrier mobility corresponds to crystalline domain size [29,30], which is consistent with our study. 

In summary, as shown in Figure 5, the device performances were significantly improved by the proposed high-K PVA/low-K PVP bilayer structure based upon the high-K characteristics of PVA and the hydrophobic surface of PVP. This led to an increased drain current and an enlarged pentacene grain size, which in turn resulted in improved performances. Figure 3 shows the optimal values of the gate capacitance to acquire the dielectric constant of 5.6 for the high-K PVA/low-K PVP bilayer structure. As shown in Figure 6, the larger contact angle of the high-K PVA/low-K PVP bilayer structure showed greater hydrophobic activity than that of the single PVA surface, which resulted in the enlarged grain sizes shown in Figure 7. We presume that the increased gate capacitance will cause an increased drain current, and the enlarged grain size will result in improved field-effect mobility. The result clearly points out that using a high-K PVA/low-K PVP bilayer enhances pentacene growth, this provides the formation of material with large grains that could potentially lead to the low presence of defects and significantly improve performances by the point of view of mobility. Nevertheless, the presence of OH ions can be reduced by tuning the proper weight percentage of PVA with respect to PVP, as shown in Figure 4.

## 4. Conclusions

Herein, we demonstrated the use of the high-K PVA/low-K PVP bilayer structure as a gate insulator of an OTFT to achieve improvements in device performance. The dielectric constant of the bilayer gate dielectric is about 5.6, which was constructed by a PVA (12 wt%) of 300 nm combined with a PVP of 500 nm. The grain size of pentacene was enlarged from 0.24 to 2.16 nm for growth on the surface of the single PVA and the bilayer high-K PVA (12 wt%)/low-K PVP, respectively. Device performances were significantly improved by use of the high-K PVA (12 wt%)/low-K PVP bilayer gate insulator, especially in the improved mobility, which is 7 times higher than that of a conventional device. We presume that the increased dielectric constant can cause increased drain current as a result of increased gate capacitance. Increased mobility is attributed to the enlarged pentacene grain size because the high-K PVA/low-K PVP bilayer layer has a more hydrophobic surface compared to the single PVP layer. It is believed that the high-K PVA/low-K PVP bilayer structure used as the gate insulator of the OTFT will lead to improved device performance.

## Figures and Tables

**Figure 1 polymers-13-03941-f001:**
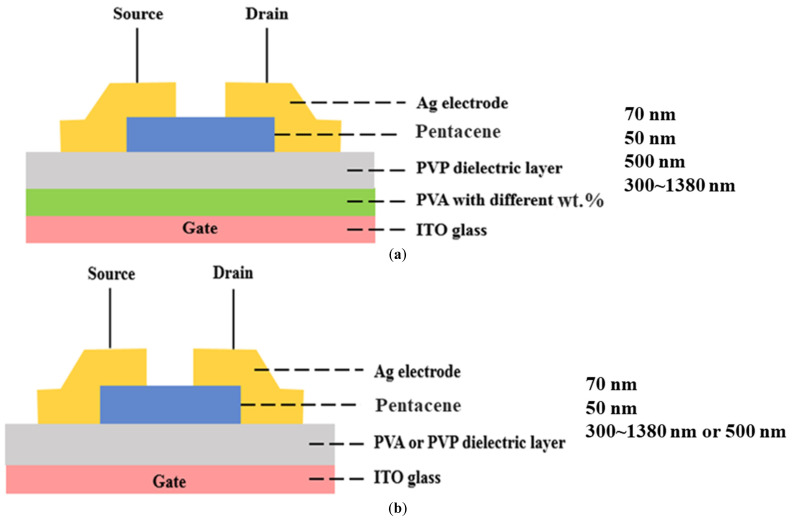
Cross-section structure of the fabricated OTFT with: (**a**) high-K PVA/low-K PVP bilayer gate dielectric; (**b**) PVA or PVP single gate dielectric.

**Figure 2 polymers-13-03941-f002:**
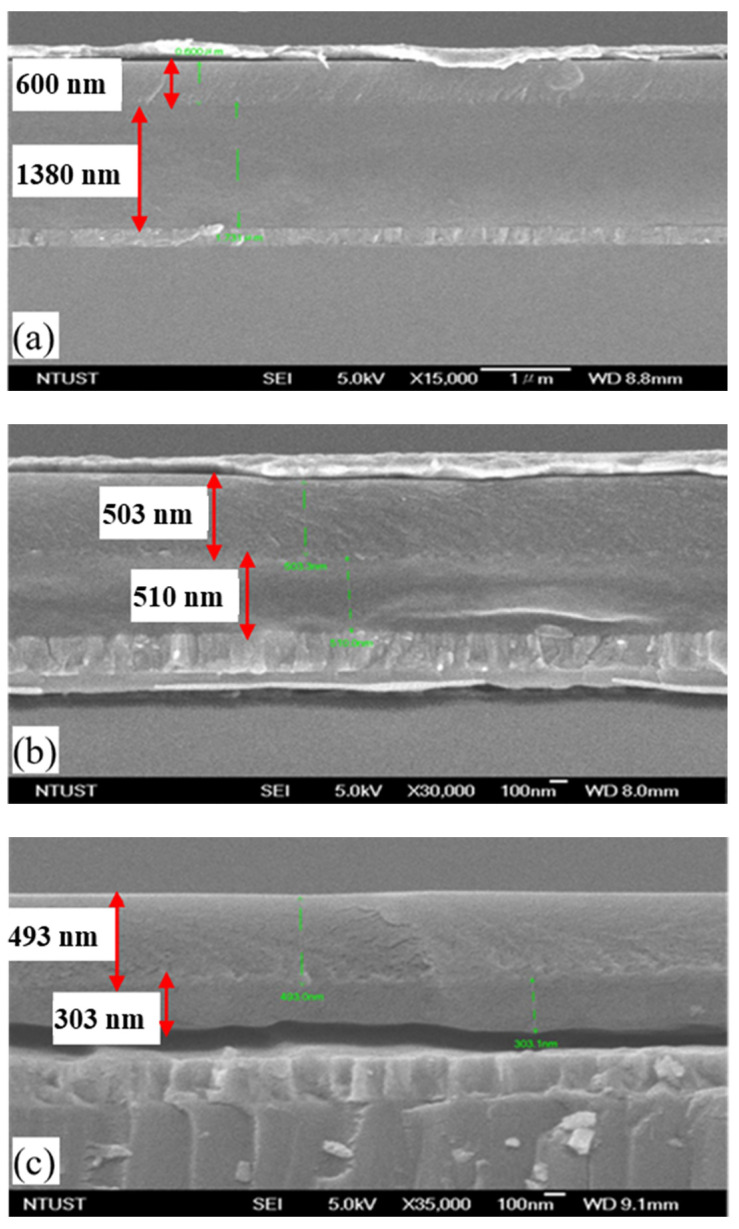
SEM images of the bilayer gate dielectric with the different PVA concentrations: (**a**) 25%, (**b**) 16%, and (**c**) 12% weight percentages and a PVP layer of 500 nm.

**Figure 3 polymers-13-03941-f003:**
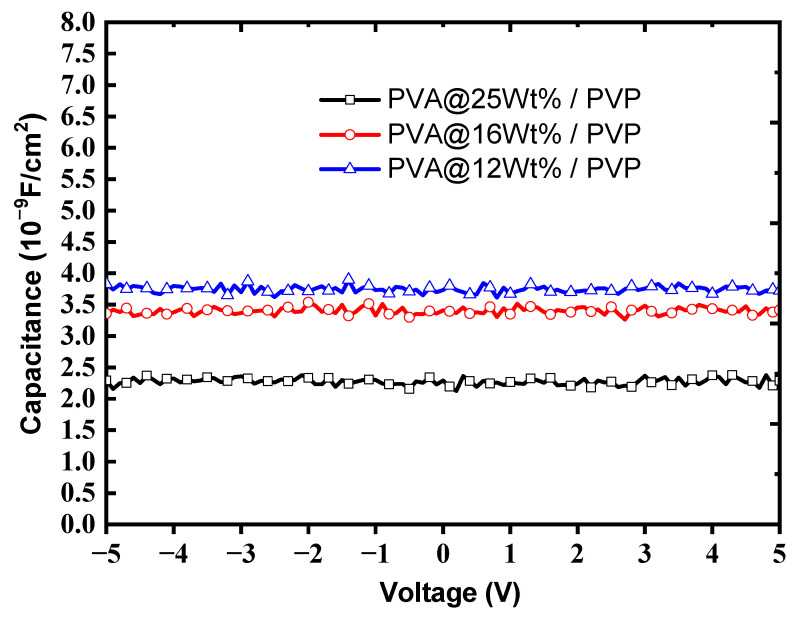
Capacitance–voltage measurement of the bilayer gate dielectric with different PVA concentrations (25%, 16%, and 12% weight percentage) and the PVP layer of 500 nm.

**Figure 4 polymers-13-03941-f004:**
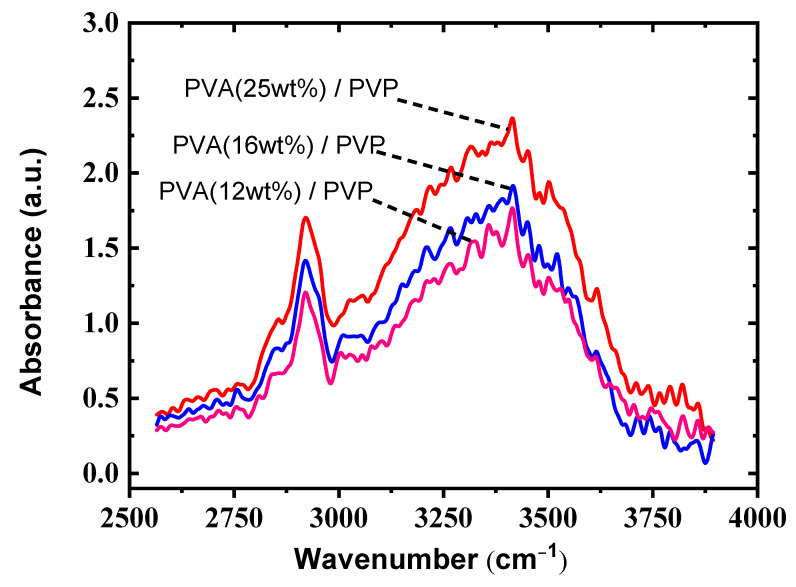
FTIR measurement of the bilayer gate dielectric with different PVA concentrations combined with a PVP of 500 nm.

**Figure 5 polymers-13-03941-f005:**
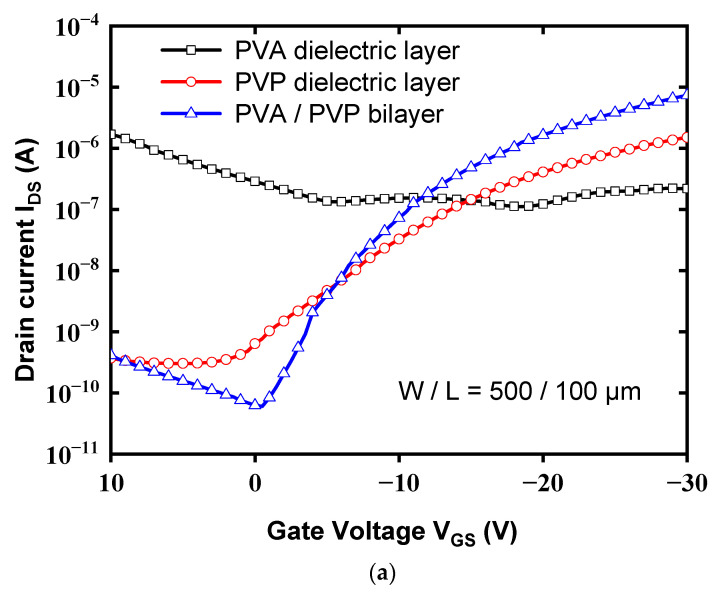
(**a**) Transfer characteristics of the devices with different gate dielectric surfaces measured at V_DS_ = −20 V. (**b**) I_GS_–V_GS_ curve of the devices with different gate dielectric surfaces measured at V_DS_ = −20 V. (**c**) Output curves of the devices with high-K PVA/low-K PVP bilayer dielectric. (**d**) Output curves of the devices with PVP dielectric layer.

**Figure 6 polymers-13-03941-f006:**
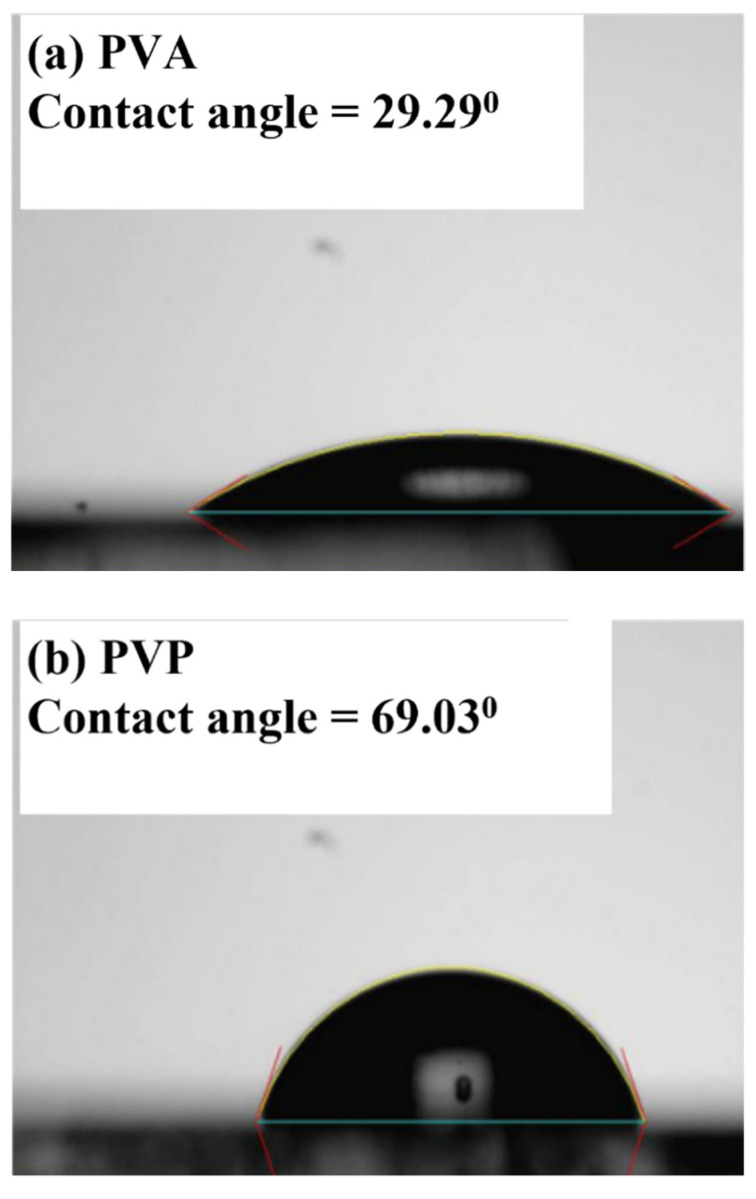
Contact angles of the different dielectric surfaces: (**a**) PVA, (**b**) PVP, and (**c**) high-K PVA/low-K PVP.

**Figure 7 polymers-13-03941-f007:**
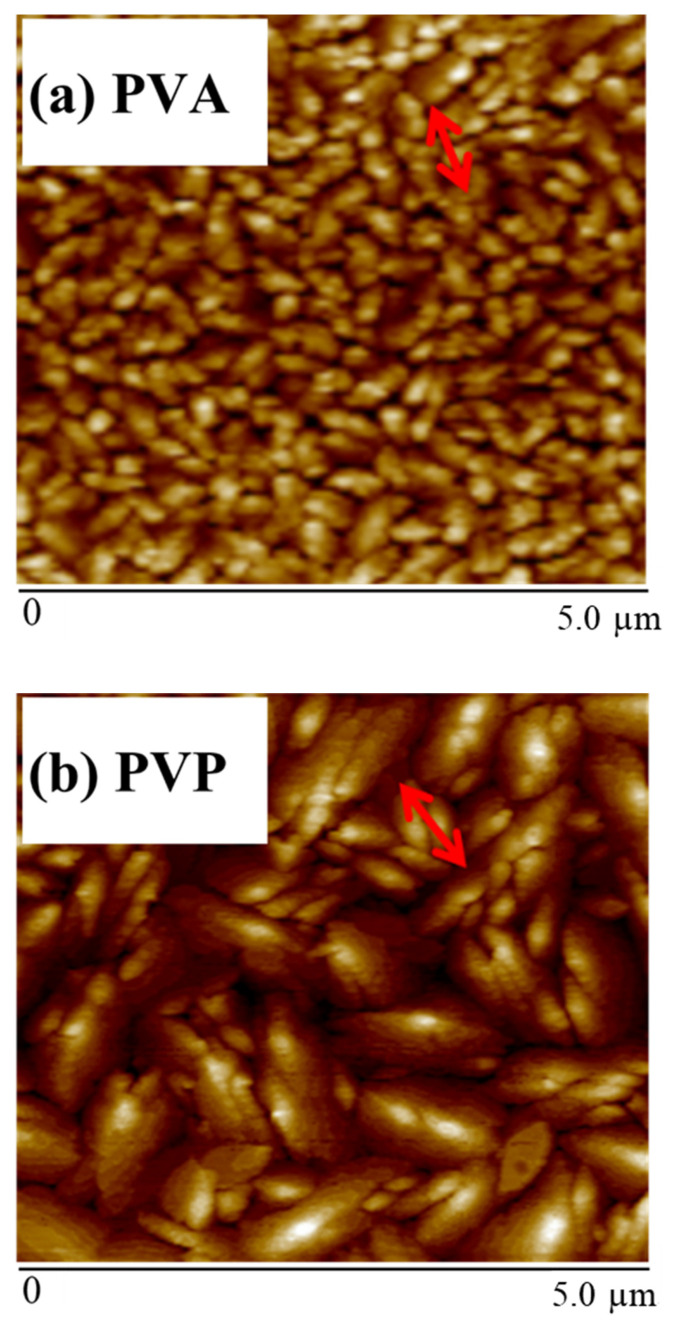
Grain size of the pentacene layer on different dielectric surfaces: (**a**) PVA, (**b**) PVP, (**c**) high-K PVA/low-K PVP. The average grain sizes are 0.24 μm, 1.58 μm, and 2.16 μm, respectively.

**Table 1 polymers-13-03941-t001:** Electrical parameters of the devices with various gate dielectrics.

Insulator Layer	V_TH_ (V)	Mobility (cm^2^/ Vs)	S.S. (V)	I_ON_/I_OFF_ Ratio
PVA	NA	NA	NA	NA
PVP	−9.4	0.16	3.94	4.99 × 10^3^
PVA/PVP	−8.6	1.12	1.41	1.21 × 10^5^

**Table 2 polymers-13-03941-t002:** Contact angles and surface energies of the different gate insulators. The dispersion and polar components of the surface energy are represented as γ^d^ and γ^p^, respectively.

	Contact Angle (°)				
Insulator Layer	DI Water	Diiodomethane	γ^d^ (mJ/m^2^)	γ^p^ (mJ/m^2^)	γ (mJ/m^2^)
PVA	29.29	22.71	46.93	25.64	72.57
PVA/PVP	66.54	26.50	45.60	7.35	52.95
PVP	69.03	29.80	44.30	6.60	50.90

## Data Availability

The data presented in this study are available on request from the corresponding author.

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
