# Peer review of "Performance Enhancement of Pentacene-Based Organic Thin-Film Transistors Using a High-K PVA/Low-K PVP Bilayer as the Gate Insulator"

_polymers, 2021, doi:10.3390/polym13223941_

Round 1

Reviewer 1 Report

The following issues must be addressed: 1. Remove the underline from rows 55-57. 2. The last paragraph from Introduction should outline what is new and innovative in this work compared with other similar papers. 3. The purity of each substance should be provided. 4. Figure 1 – remove the red underlines 5. All the experimental discussions are based on observations. However, thi is not enough. The authors should provide deep interpretations and correlations between results. 6. Is not clear why parts of the main text are underlined. 7. Table 1- the units for mobility must be corrected. 8. The polar and dispersive component of the surface energy should be provided. Overall, some parts of the manuscript seem to be written in a hurry. Please spend more time on evaluating the significance of each result.

Author Response

Response to Reviewer 1

The following issues must be addressed:

Point 1: Remove the underline from rows 55-57.

Response 1:

Thank you for your kindly comment. We had deleted the underline from rows 55-57 in according with the reviewer suggestion.

Point 2: The last paragraph from Introduction should outline what is new and innovative in this work compared with other similar papers.

Response 2:

Thank you for your kindly comment. We had added the similar papers as references and the related descriptions in the revised manuscript on page 2, line 62 to line 71.

Detail description as below:

Compared with other similar papers, the improved uFE in our study is about 1.12cm2/Vs, significantly better than that of the reported papers previously [19]-[22]. The obvious performance improvement can be attributed to the high-K PVA/low-K PVP bilayer structure based upon the high-K characteristics of PVA and the hydrophobic surface of PVP; this led to an increased drain current and an enlarged pentacene grain size, which in turn resulted in improved performances. Thus, it is believed that the proposed high-K PVA/low-K PVP structure is a good candidate for the performance improvement because it can not only improve the device performances but also provide the advantages of the simple process, low cost and the avoiding of the cross-linking process of PVA using toxic agents, in comparison with the similar reports [17]-[22].

Point 3: The purity of each substance should be provided.

Response 3:

Thank you for your kindly comment. We had added the related descriptions in the revised manuscript on page 2, line 74, line 77, line 82 and line 83.

  1. The composition of PVP material is PVP: PMCF: PGMEA= 2:1: 20
  2. PVA (molecular weight = 46,000–186,000)
  3. ITO substrate: The resistivity is about 20~40 Ω‧cm
  4. Pentacene materials: Aldrich Chem. Co., 99% purity

Point 4: Figure 1 – remove the red underlines

Response 4:

Thank you for your kindly comment. We had removed the red underline in the figure 1 in according with the reviewer suggestion.

Point 5: All the experimental discussions are based on observations. However, this is not enough. The authors should provide deep interpretations and correlations between results.

Response 5:

We had added the related descriptions in the revised manuscript on page 9, from line 203 to line  217 and from page 12, line 249 to page 13, line 263.

Detail description as below:

  1. page 9, from line 203 to line 217: The descriptions are the same as the response 8
  2. page 12, line 249 to page 13, line 26: In summary, as shown in figure 5, the device performances had been significantly improved by the proposed high-K PVA/low-K PVP bilayer structure based upon the high-K characteristics of PVA and the hydrophobic surface of PVP; this led to an increased drain current and an enlarged pentacene grain size, which in turn resulted in improved performances. The figure 3 shows the optimal values of the gate capacitance to acquire the dielectric constant of 5.6 for the high-K PVA/low-K PVP bilayer structure. As shown in figure 6, the larger contact angle of high-K PVA/low-K PVP bilayer structure showed greater hydrophobic activity than that of single PVA surface, which result in the enlarged grain sizes shown in figure 7. We presume that the increased gate capacitance will cause the increased drain current, and the enlarged grain size will result in the improved field-effect mobility. The result clearly points out that using high-K PVA/low-K PVP bilayer enhances pentacene growth, this provides the formation of material with large grains that could potentially lead to low presence of defects and significantly improve performances by the point of view of mobility. Nevertheless, the presence of OH ions can be reduced by tuning the proper weight percentage of PVA with respect to PVP, as shown in figure 4.

Point 6: Is not clear why parts of the main text are underlined.

Response 6:

Thank you for your kindly comment and sorry for your confusion. We had deleted all the underline mark in the revised manuscript.

Point 7: Table 1- the units for mobility must be corrected.

Response 7: Thank you for your kindly comment. We had corrected the units for mobility in Table 1 in the revised manuscript.

Table 1. The electrical parameters of the devices with various gate dielectrics.

Insulator layer

VTH (V)

Mobility (cm2/ Vs)

S.S. (V)

ION / IOFF ratio

PVA

NA

NA

NA

NA

PVP

-9.4

0.16

3.94

4.99x103

PVA/PVP

-8.6

1.12

1.41

1.21x105

Point 8: The polar and dispersive component of the surface energy should be provided.

Response 8:

Thank you for your kindly comment. We had added the polar and dispersive component of the surface energy in the table II.  And we also added the related descriptions in the revised manuscript on page 9, line 203 to line 217.

Detail description as below:

Table II shows the contact angles of DI water and diiodomethane and the surface energies of gate insulators. The γd and γp represent dispersion and polar components of the surface energy (γ), respectively. It shows the surface energy values, 72.57 mJ/cm2, 52.95 mJ/cm2, and 50.90 mJ/cm2 for High-K PVA, High-K PVA/Low-K PVP, and Low-K PVP, respectively. The decreased surface energy of the PVA/PVP bilayer are comparable to that of the PVP layer due to the removal of –OH groups from the PVP by the cross-linking process. It is noted that the surface energy of PVA is higher than that of PVA/PVP or PVP layers due to lots of –OH groups. It is also consistent with the measured contact angles in the fig. 6 because the smaller contact angle represents the higher surface energy. The smaller contact angle and higher surface energy of PVA show the hydrophilic surface with OH groups, which could result in a stronger interaction between the dielectric surface and the pentacene molecules during the deposition to inhibits the growth of the pentacene grain. In contrast, pentacene exhibits large grains when grown on hydrophobic insulator surface with high contact angle and low surface energy.

 Table II. Contact angles and surface energies of the different gate insulators. The dispersion and polar components of the surface energy represented as γd and γp, respectively.

Contact angle (0)

Insulator layer

DI water

Diiodomethane

γd

(mJ/m2)

γp

(mJ/m2)

γ

(mJ/m2)

PVA

29.29

22.71

46.93

25.64

72.57

PVA/PVP

66.54

26.50

45.60

7.35

52.95

PVP

69.03

29.80

44.30

6.60

50.90

Reviewer 2 Report

This proposal paper is a helpful contribution to the specific implementation of the (high-K PVA/low-K PVP) 'PVA/PVP gate insulator bilayer', since it enhances the performance of the 
(Pentacene-based) Organic Thin Film Transistors, tested under (common) actual bias stress conditions. 

However, there are (4) minor notes/corrections:
1. Line_013: A (term) notation correction/refinement (do the same, in: Line_181). Remake/refine the term 'dielectric constant' used for a (pure/single) material, for a bilayer to be 'based organic thin film transistor. The dielectric constant of the optimal PVA/PVP bilayer is 5.6,".

2. Line_020: Some (units) notation correction (do the same, in: Line_121). Remake the unit 'cm2/V∙sec',  to be 'cm2/(V s)', in: "creased from 0.16 to 1.12 cm2/V∙sec, 7 times higher than that of the control sample.". 

3. Line_113: Add a reference for (/after) the statement, 'induces complexity' (for better tracing and clarity), in: "tion through the thickness reduction process and then induces complexity to the device.".

4. Line_128: A wording refinement. Remake/refine the word 'dielectric' (capacitance) to be 'actual' or 'effective' (capacitance), in: "proposed PVA/PVP bilayer structure could obviously increase the dielectric capacitance". 

Author Response

Response to Reviewer 2

This proposal paper is a helpful contribution to the specific implementation of the (high-K PVA/low-K PVP) 'PVA/PVP gate insulator bilayer', since it enhances the performance of the 
(Pentacene-based) Organic Thin Film Transistors, tested under (common) actual bias stress conditions. 

However, there are (4) minor notes/corrections:

Point1: Line_013: A (term) notation correction/refinement (do the same, in: Line_181). Remake/refine the term 'dielectric constant' used for a (pure/single) material, for a bilayer to be 'based organic thin film transistor. The dielectric constant of the optimal PVA/PVP bilayer is 5.6,".

Response 1:

Thank you for your kindly comment. We take your suggestion to correct the term (PVA/PVP) to high-K PVA/low-K PVP. Besides, we also had corrected all the PVA/PVP term changing to high-K PVA/low-K PVP in the revised manuscript.

Point 2: Line_020: Some (units) notation correction (do the same, in: Line_121). Remake the unit 'cm2/V∙sec', to be 'cm2/(V s)', in: "creased from 0.16 to 1.12 cm2/V∙sec, 7 times higher than that of the control sample.".

Response 2:

Thank you for your kindly comment. We take your suggestion to refine the unit 'cm2/V∙sec' to 'cm2/(Vs)' in the revised manuscript.

Point 3:  Line_113: Add a reference for (/after) the statement, 'induces complexity' (for better tracing and clarity), in: "tion through the thickness reduction process and then induces complexity to the device.".

Response 3:

Thank you for your kindly comment. We had added the reference as:

[23] V. Singh, B. Mazhari, “Impact of scaling of dielectric thickness on mobility in top-contact pentacene organic thin film transistors,” Journal of Applied Physics 111, 034905 (2012).

Point 4:  Line_128: A wording refinement. Remake/refine the word 'dielectric' (capacitance) to be 'actual' or 'effective' (capacitance), in: "proposed PVA/PVP bilayer structure could obviously increase the dielectric capacitance".

Response 4:

Thank you for your kindly comment. By the reviewers’ suggestion, we had corrected the wording refinement on page 4, line 131 in the revised manuscript.

Round 2

Reviewer 1 Report

The manuscript can be published in present form.